# A Smart Procedure for Assessing the Health Status of Terrestrial Habitats in Protected Areas: The Case of the Natura 2000 Ecological Network in Basilicata (Southern Italy)

Vito Imbrenda, Maria Lanfredi, Rosa Coluzzi and Tiziana Simoniello *

Institute of Methodologies for Environmental Analysis (IMAA), Italian National Research Council (CNR), Tito Scalo, 85050 Potenza, Italy; vito.imbrenda@imaa.cnr.it (V.I.); maria.lanfredi@imaa.cnr.it (M.L.); rosa.coluzzi@imaa.cnr.it (R.C.)
* Correspondence: tiziana.simoniello@imaa.cnr.it

**Abstract:** Natura 2000 is the largest coordinated network of protected areas in the world, which has been established to preserve rare habitats and threatened species at the European Community level. Generally, tools for habitat quality assessment are based on the analyses of land-use/land-cover changes, thus, highlighting already overt habitat modifications. To evaluate the general quality conditions of terrestrial habitats and detect habitat degradation processes at an early stage, a direct and cost-effective procedure based on satellite imagery (Landsat data) and GIS (Geographic Information System) tools is proposed. It focuses on the detection of anomalies in vegetation matrix (stress/fragmentation), estimated for each habitat at the level of both a single protected site and local network, to identify habitat priority areas (HPA), i.e., areas needing priority interventions, and to support a rational use of resources (field surveys, recovery actions). By analyzing the statistical distributions of standardized NDVI for all the enclosed habitats (at the site or network level), the Degree of Habitat Consistency (DHC) was also defined. The index allows the assessment of the general status of a protected site/network, and the comparison of the environmental conditions of a certain habitat within a given protected site (SCI, SAC) with those belonging to the other sites of the network. The procedure was tested over the Natura 2000 network of the Basilicata region (Southern Italy), considered as a hotspot of great natural and landscape interest. An overall accuracy of ~97% was obtained, with quite low percentages of commission (~8%) and omission (~6%) errors. By examining the diachronic evolution (1985–2009) of DHC and HPA, it was possible to track progress or degradation of the analyzed areas over time and to recognize the efficaciousness/failure of past managements and interventions (e.g., controlled disturbances), providing decision-makers with a thorough understanding for setting up the most suitable mitigation/contrast measures.

**Keywords:** Natura 2000; habitat conservation; controlled disturbance; landsat; NDVI; land degradation; Southern Italy

## 1. Introduction

Human activities and climate change are driving significant modifications in terrestrial ecosystems, often implying the impoverishment of the floristic and faunal composition of natural habitats [1]. Several international initiatives have been launched worldwide over the last few decades to prevent biodiversity loss; among them, the European Natura 2000 network represents the largest coordinated network of protected sites. It has been established on the basis of the Habitats and Birds Directives (92/43/EC and 2009/147/EC, respectively) to preserve threatened species within their habitats and to guarantee the long-term survival of ecological functions and services [2,3].

Within Natura 2000 network, each Member State is called upon to ensure close monitoring of its sites through a periodic reporting (every six years) to pursue conservation

objectives. For such periodic monitoring, the European Commission has explicitly suggested the use of Earth Observations (EO) techniques to develop continuous, reliable, transferable, and standardized procedures [4].

Optical remotely sensed data have been historically used to map land cover as a basis for habitat mapping, and biodiversity assessment in general, by analyzing broad habitat types (forests, grasslands, wetlands, etc.) at different scales (from quite coarse 250 m–1 km of MODIS-Moderate Resolution Imaging Spectroradiometer to a few meters of Landsat (30 m), Aster (15 m), Ikonos (2.62 m)) (see, e.g., [5–9]). The availability of Sentinel-2 data in the last seven years has promoted a significant production of studies and projects to exploit the sensor characteristics (spatial and temporal resolution) for the mapping and monitoring of Natura 2000 habitats by implementing integrated or new classification approaches [10–13].

To evaluate the status of habitats, many recent studies are based on the assessment of Land Cover/Land Use (LC/LU) changes [12,14]. Additionally, the two Copernicus products (N2K and N2K change, https://land.copernicus.eu/local/natura, last accessed on 30 March 2022) recently released (July 2021) for biodiversity monitoring in Natura 2000 sites are essentially based on LU/LC. A disadvantage of such an approach is that changes are identified when they are already evident (thus, variations that have already led to a modification of the habitat with the relative consequences for the hosted species) or the assessment of potential threats and pressures of surrounding areas (i.e., potential habitat deterioration/loss).

The development of lightweight hyperspectral and multispectral sensors for unmanned aerial vehicles (UAVs) has supported advances in identifying natural and semi-natural habitats at a scale more suitable for the Natura 2000 network by taking into account species-level classification, thus providing the ability to map even complex habitats, such as those of wetland areas as standalone tool [15] or combined with satellite imagery [16]. The coverage of large areas, such as networks of protected areas, collides with the well-recognized drawbacks of such systems: costs and implementation times and data intercalibration.

Among free downloadable missions, Landsat is the only one capable of providing data over a period of time consistent with the Natura 2000 establishment and with a good compromise in terms of spectral and spatial resolution (e.g., [17–19]). Moreover, the recent launch of Landsat 9 (27 September 2021) ensured the continuity of the Landsat program beyond 2030.

The assessment of habitat quality with remote tools is complex due to the complexity of factors and mechanisms that determine the health of habitats [20–23]. One of the main indicators of habitat quality is the status of vegetation cover in terms of biomass amount, canopy vigor, phenology, and distribution of plant species [24–26]. From this perspective, the IPBES has produced a thematic assessment of land degradation and restoration, based on the evaluation of vegetation conditions, to enhance the knowledge base for policy makers [27–30].

To assess the actual habitat status and detect signs of habitat degradation at an early stage, the proposed smart procedure is based on the statistical distributions of standardized NDVI (Normalized Difference Vegetation Index) that can be easily implemented with the support of GIS (Geographic Information System) tools, even for non-experts in remote sensing. NDVI represents the most used indicator of the state of vegetation (density and photosynthetic activity) and can be estimated by a wide range of optical sensors that acquire in the red and NIR bands; see, for example, [13,16,31–35].

The final goal of this procedure is to obtain a spatially explicit assessment of the habitat conditions, evaluated both at the protected site and local network levels, localizing those areas that are suspected of incoming degradation (HPA—Habitat Priority Areas) and evaluating the overall habitat status (DHC—Degree of Habitat Consistency). These satellite-based products can play an important guiding role for managers, as they are intended to supply fast screenings of large areas with simple tools (soft assessment), enabling the

identification of anomalous portions of habitats that require additional in-field tailored investigations (hard assessment). The relatively speedy procedure and its low cost allow for a close monitoring of protected sites and the early detection of priority areas to be inserted into intervention plans [36,37].

The present work, supported by the precious availability of in-field data for the period 2009–2010, shows the efficaciousness of the proposed procedure with a diachronic study (1985–2009) on the Natura 2000 network of Basilicata (Southern Italy), which is recognized as a hotspot of great natural interest [38,39] under the threat of increasing anthropogenic pressure [40–43].

## 2. Materials and Methods

### 2.1. Study Area

#### 2.1.1. Basilicata Region

The investigated area includes some of the protected sites of the Basilicata Natura 2000 network (Southern Italy, Figure 1), which is characterized by a large variety of landscapes and climatic features. Orography and distance from the sea are the main factors affecting the extent and type of natural vegetation. Dense forests, grasses, and pastures are located in mountainous areas (Apennine chain culminating in the Pollino Massif); maquis and sparsely vegetated areas dominate in the central-eastern part of the region. Moreover, anthropic covers devoted to agriculture are prevalently located in lowland areas (orchards along the Ionian coast, vineyards and olive groves in the volcanogenic Vulture basin, and monoculture of cereals in the north-eastern part of the region; see, e.g., [44–47]). The strong biogeographic heterogeneity of the area follows the local climate spatial variability encompassing: the typical Mediterranean regime (along coastlines), with hot summers and cold, rainy winters; the mountain climate (cold, wet, and often snowy); the subcontinental one (in the inner and hilly areas) with very hot and dry summers, mild winters, and a low amount of precipitation [48–50].

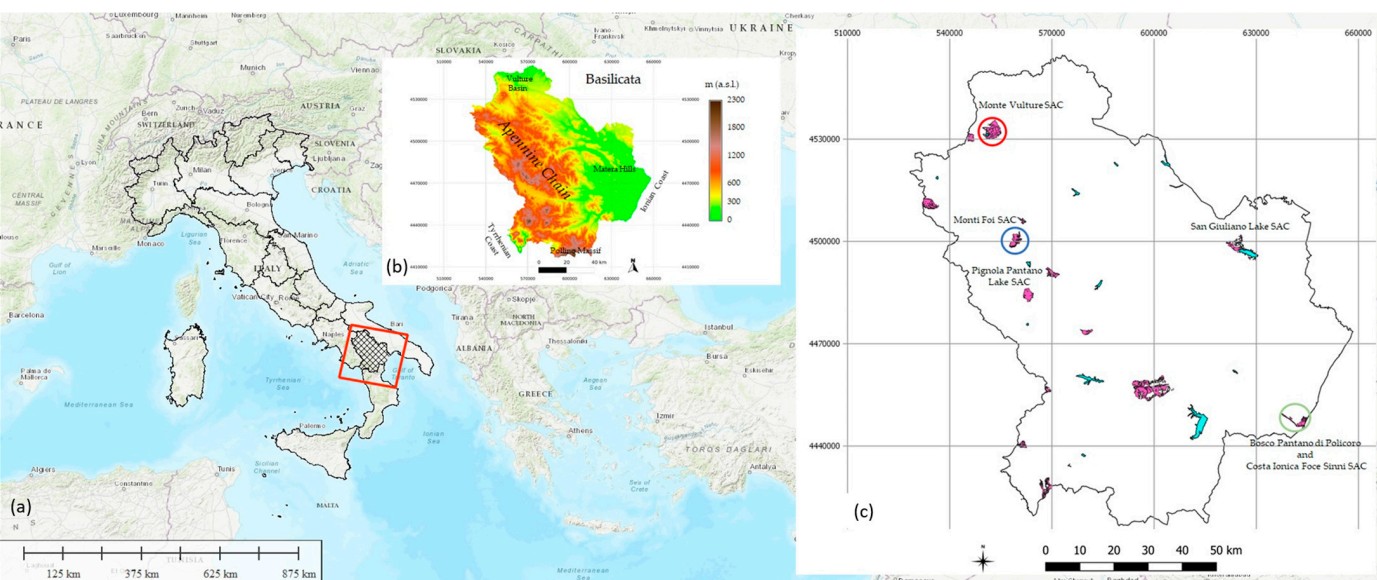

**Figure 1.** (**a**) Location of Basilicata with the 20 administrative units (NUTS2—Nomenclature of territorial units for statistics) of Italy and in red the footprint of Landsat path 188 and row 032; (**b**) DEM (Digital Elevation Model) of the study area with the main natural toponyms; (**c**) investigated SACs of Natura 2000 network of the Basilicata. Sites discussed in depth as case studies are circled, whereas in cyan, the main inner water bodies are reported.

Basilicata is among the most affected Italian regions in terms of environmental risks (land degradation, landslides, floods, etc.; see, e.g., [51–58]).

2.1.2. The Natura 2000 Network of the Basilicata Region

Natura 2000 is an European ecological network composed of sites designated under the EU Birds Directive (Special Protection Areas) and the Habitats Directive (Sites of Community Importance and Special Areas of Conservation) following the listed criteria:

- Special Protection Area (SPA): 1% of the population of listed vulnerable species or wetlands of international importance for migratory waterfowl;
- Sites of Community Importance (SCI): habitat types listed in the directive's Annex I and the habitats of the species listed in its Annex II;
- Special Areas of Conservation (SAC): priority SCI sites that are most threatened and/or most important for conservation where the conservation necessary measures have been planned for the maintenance or restoration of natural habitats and peculiar species.

In this study, 16 Sites of Community Importance (SCI) of the Basilicata Natura 2000 network that have become Special Areas of Conservation (SAC) in 2013 (in accordance with the Ministerial Decree of 21 February 2013) were analyzed. They cover a surface of about 11,350 ha (Figure 1c). Such sites have been included in the first phase of regional monitoring plans providing for extensive field surveys.

*2.2. Data*

2.2.1. Natura 2000 Habitat Boundaries

The analyses were implemented at two levels: at the network and site level. In the first case, the status of a given habitat was assessed with respect to all 16 sites shown in Figure 1c (where the habitat is present), thus highlighting any priority of investigation and detailed intervention among the protected sites (network-level priority). At the site level, analysis is restricted to a single SAC and the critical portions of a considered habitat within each protected area were analyzed, highlighting any priorities within the site itself (site-level priority). Basically, from a management point of view, with these two levels of analyses, the method can support local managing bodies and the regional authority by identifying priority areas at site (local) or network (regional) level.

To present the site-level analyses, three protected areas with the highest number of habitats and representing the wide biogeographical gradient of the Basilicata landscapes were selected: they are representative of hilly (IT9210210), mountainous (IT9210215), and coastal plain (IT9220055) environments.

- Monte Vulture

One of the most valuable SACs in Basilicata is the Monte Vulture (IT9210210), characterized by eight different habitats (Table 1). The site was proposed as SCI in 1995, confirmed in 2006, and then designated as a SAC in 2013. It was also classified as a SPA in 1999.

**Table 1.** Habitats of the examined SACs and their corresponding areas (in hectares and percentage on total site surface).

| SAC | Habitat Code | Description | Area (ha) | Area (%) |
|---|---|---|---|---|
| | 3150 | *Natural eutrophic lakes with Magnopotamion or Hydrocharition-type vegetation* | 27.5 | 1.7 |
| | 6420 | *Mediterranean tall humid herb grasslands of the Molinio-Holoschoenion* | 7.5 | 0.5 |
| | 91B0 | *Thermophilous Fraxinus angustifolia woods* | 3.4 | 0.2 |
| Monte Vulture | 91E0* | *Alluvial forests with Alnus glutinosa and Fraxinus excelsior* | 21.9 | 1.4 |
| IT9210210 | 91M0 | *Pannonian-Balkanic turkey oak–sessile oak forests* | 451.5 | 28.8 |
| | 9220* | *Apennine beech forests with Abies alba and beech forests with Abies nebrodensis* | 290.4 | 18.5 |
| | 9260 | *Castanea sativa woods* | 693.1 | 44.2 |
| | 9510* | *Southern Apennine Abies alba* | 74.0 | 4.7 |
| | 3150 | *Natural eutrophic lakes with Magnopotamion or Hydrocharition-type vegetation* | 0.13 | 0.02 |
| | 3260 | *Water courses of plain to montane levels with the Ranunculion fluitantis and Callitricho-Batrachion vegetation* | 0.10 | 0.01 |
| | 6210 | *Semi-natural dry grasslands and scrubland facies on calcareous substrates* | 6.82 | 0.85 |
| | 6210* | *Semi-natural dry grasslands and scrubland facies on calcareous substrates (*important orchid sites)* | 71.49 | 8.88 |

**Table 1.** *Cont.*

| SAC | Habitat Code | Description | Area (ha) | Area (%) |
|---|---|---|---|---|
| Monti Li Foi | 6210,6430 | *Semi-natural dry grasslands and scrubland facies on calcareous substrates/Hydrophilous tall herb fringe communities of plains and of the montane to alpine levels* | 9.17 | 1.14 |
| IT9210215 | 6430 | *Hydrophilous tall herb fringe communities of plains and of the montane to alpine levels* | 1.32 | 0.16 |
| | 6510 | *Lowland hay meadows* | 44.15 | 5.48 |
| | 8130, 8220 | *Western Mediterranean and thermophilous scree/ Siliceous rocky slopes with chasmophytic vegetation* | 14.05 | 1.75 |
| | 9180* | *Tilio-Acerion forests of slopes, screes and ravinen* | 2.70 | 0.34 |
| | 91M0 | *Pannonian-Balkanic turkey oak–sessile oak forests* | 131.41 | 16.32 |
| | 9210 | *Apennine beech forests with Taxus and Ilex* | 523.9 | 65.06 |
| | 1130 | *Estuaries* | 3.8 | 0.6 |
| | 1210, 2110, 2120, 2210, 2230, 2240 | *Different Dune Habitats* | 14.7 | 2.4 |
| | 1310, 1420 | *Salicornia and other annuals colonizing mud and sand, Mediterranean and thermo-Atlantic halophilous scrubs* | 12.6 | 2.0 |
| Bosco Pantano di Policoro IT9220055 | 1410, 6420 | *Mediterranean salt meadows, Mediterranean tall humid herb grasslands of the Molinio-Holoschoenion* | 55.3 | 8.9 |
| | 1410, 92D0 | *Mediterranean salt meadows, Southern riparian galleries and thickets* | 154.9 | 24.9 |
| | 2250*, 2260, 2230 | *Coastal dunes with Juniperus spp, Cisto-Lavanduletalia dune sclerophyllous scrubs, Malcolmietalia dune grasslands* | 31.9 | 5.1 |
| | 2260 | *Cisto-Lavanduletalia dune sclerophyllous scrubs* | 49.3 | 7.9 |
| | 3280 | *Constantly flowing Mediterranean rivers with Paspalo-Agrostidion species and hanging curtains of Salix and Populus alba* | 6.7 | 1.1 |
| | 91F0 | *Riparian mixed forests of Quercus robur, Ulmus laevis, and Ulmus minor, Fraxinus excelsior, or Fraxinus angustifolia, along the great rivers* | 144.1 | 23.2 |
| | 91F0, 92A0 | *Fraxinus angustifolia, along the great rivers, Salix alba and Populus alba galleries* | 53.6 | 8.6 |
| | 92D0 | *Southern riparian galleries and thickets* | 95.4 | 15.3 |

The Monte Vulture site has been recently included within the homonymous Regional Park established by the Basilicata Region (November 2017) because of its unique concentration of geological features (e.g., the volcanic lakes of Monticchio) and floro-faunistic assemblages. It represents a great attraction for hit-and-run tourism practiced throughout the year, threatening the naturalness of the site. In addition to this, there are important industrial activities involved in the exploitation of the vast groundwater reserves of the SAC to produce mineral waters.

Details on Natura 2000 standard data form can be found on the EEA platform at https://natura2000.eea.europa.eu/Natura2000/SDF.aspx?site=IT9210210 (last accessed on 30 March 2022).

- Monti Li Foi

The second examined SAC is the Monti Li Foi (IT9210215), composed of eleven different habitats (Table 1) located at high altitude (1100–1350 m a.s.l.) in a landscape rich in floristic composition and still relatively low-impacted by tourism flows. The site was proposed as SCI in 1995, confirmed in 2006, and then designated as SAC in 2013.

Details on the Natura 2000 standard data form can be found on the EEA platform at https://natura2000.eea.europa.eu/Natura2000/SDF.aspx?site=IT9210215 (last accessed on 30 March 2022).

- Bosco Pantano di Policoro

Basilicata encompasses some important protected sites located along the Ionian coastline. Among them is the SAC of Bosco Pantano di Policoro and Costa Ionica Foce Sinni (IT9220055). The site was classified as SCI in 1995 and then proposed as SAC in 2013; subsequently, it was extended by adding to the original terrestrial habitats, the marine habitats of the overlooking coast and, finally, it was designated as whole SAC in 2017. It was also classified as SPA in 1999. The Bosco Pantano site includes a residual strip (about 50 hectares) of what, until a few decades ago, was one of the largest lowland forests in southern Italy (550 hectares in 1971). Inside it is the "big oak" of Bosco Pantano di Policoro,

an example of "quercus robur" inserted for its botanical rarity (shape and bearing) among the monumental trees of Italy in the latest update of the official list of the Ministry of Agricultural and Forestry Policies, published by DIFOR (Italian General Management for Mountain Economy and Forests) on 5 May 2021. The analyses were implemented only on the 11 terrestrial habitats (Table 1). The site also contains a WWF Oasis and represents a very interesting case of how anthropogenic forces have contributed to shape the physical environment through changes in soil use and vegetation pattern.

Details on Natura 2000 standard data form can be found on the EEA platform at https://natura2000.eea.europa.eu/Natura2000/SDF.aspx?site=IT9220055#7 (last accessed on 30 March 2022).

### 2.2.2. Satellite Data

To analyze habitat conditions of the Natura 2000 network in the Basilicata region, data from the Landsat mission were analyzed. For the Basilicata region, a single tile covers all the selected Natura 2000 sites; therefore, two clear-sky images of the Thematic Mapper (TM) sensor flying on the Landsat 5 satellite (188 path and 32 row GTCE—Ground Terrain Corrected Enhanced) acquired in the summer period (10 August 1985 and 27 July 2009) were freely downloaded from the USGS (United States Geological Survey) Landsat archive (http://earthexplorer.usgs.gov/, last accessed on 1 June 2021).

The 27 July 2009 scene was selected because it is contemporary with most of the field surveys and also has a very low percentage of cloud cover. The 1985 image was chosen as the best clear sky before the definition of Natura 2000 sites and useful for a multitemporal study (1985–2009).

The images were preprocessed by transforming the digital numbers into radiance units and then into reflectance according to the NASA-GSCF calibration coefficients [59,60] in PCI Geomatics. To reduce geometric inconsistency between the two selected dates, the 2009 image was used as reference to register the 1985 scene, so that a root mean square error (RMSE) less than 0.5 pixel was obtained.

### 2.2.3. Field Data

The field surveys were performed between 2009 and 2010 in all the investigated sites. The monitoring campaign, implemented by the Basilicata Region (Resolution no. 1214/2009 of the Basilicata Regional Government), was aimed at updating information on the habitat features and species, including potential widening and revision of the SCI (today SAC) boundaries. Information gathered during in situ surveys concerned different aspects of habitat structure and composition (e.g., inventory and status of flora and fauna, location and typology of anthropogenic stressors, emergence of valuable points of interest) accompanied by some ancillary data (concerning climatic characterization, geological setting, and economic and demographic status of the involved municipalities). This collection of data was used to populate the Database proposed by the Ministry of the Environment and Protection of Land and Sea of Italy for updating Natura 2000 data in accordance with Article 17 of the Habitat Directive and constitutes the basis for the drafting of protection and conservation measures and management plans. For the purpose of this study, data, reports, and photographs acquired during this extensive field campaign were used to interpret results and validate the proposed satellite-based approach.

### 2.2.4. Auxiliary Data

Freely accessible orthophotos (1:10,000) with a resolution of less than 1 m were used as reference for changes; they are available as WMS layers in GIS environment for the years 1988, 1994, 2006, 2008, 2011, and 2013. They are provided by the AGEA (Italian Agency for the Delivery in Agriculture) and the MATTM (Ministry of the Environment and Protection of Land and Sea of Italy, now replaced by the Ministry of the Ecological Transition). They are also available through the regional spatial data infrastructure of the Basilicata region

(http://rsdi.regione.basilicata.it/, last accessed on 1 June 2021). Such data were used to interpret and validate the outcomes of the proposed procedure.

*2.3. Methods*

2.3.1. Degree of Habitat Consistency (DHC) and Habitat Priority Areas (HPA)

A correct management of protected sites requires close monitoring. If monitoring is sufficiently continuous, it can be assumed that possible degradation processes are detectable at an early stage in the form of a few anomalies within a generally healthy habitat.

In this work, the spatial and temporal variability of the NDVI [61] was analyzed as proxy for vegetation density and health [62–64]:

$$\text{NDVI} = \frac{\rho_{\text{nir}} - \rho_{\text{red}}}{\rho_{\text{nir}} + \rho_{\text{red}}} \qquad (1)$$

where $\rho_{\text{nir}}$ and $\rho_{\text{red}}$ are the reflectance in the near infrared and red channels (bands 4 and 3 for the used TM sensor), respectively.

By looking at the statistics of the Standardized NDVI per habitat, the emergence of threatened areas in homogeneously healthy habitats should imply the appearance of outliers on the left tail of the standardized distribution. The work hypothesis is that the NDVI statistics of an observational sample including incoming degradation can be divided in 'outlying' and 'inlying' observations [65] and that left-tail (negative outliers) indicates the existence of mechanisms that are threatening the vegetation matrix. Though right-tail outliers are generally not directly related to degradation, they nonetheless indicate some degree of habitat alterations (alien plant invasion, encroachment processes), which should be monitored too.

In order to separate outliers from inliers areas, a simple and consolidated statistical procedure was used for identifying the so-called inner and outer fences of the statistical distribution of NDVI [66].

$$\text{LW} = Q_1 - 1.5(Q_3 - Q_1) \qquad (2)$$

$$\text{UW} = Q_3 + 1.5(Q_3 - Q_1) \qquad (3)$$

where $Q_1$, $Q_2$, and $Q_3$ are the first, the second, and the third quartile of the distribution. Values lower than LW are negative outliers, values higher than UW are positive outliers. The mapping of negative outliers identifies the areas where the vegetation matrix is threatened (stressed/fragmented) thus requiring more urgent conservation interventions (Habitat Priority Areas—HPA). Values located between LW and UW are inliers, and the index DHC (Degree of Habitat Consistency) was defined as the percentage area showing homogeneously consistent conditions, i.e., the percentage of areas of these inliers that are characterized by standardized NDVI values belonging to the same statistical distribution with good statistical confidence.

In detail, the two layers can be derived by following the flowchart of the procedure shown in Figure 2.

The principal steps of the methodology are:

- Estimation of mean values $\mu_h(\cdot)$ and standard deviation $\sigma_h(\cdot)$ of NDVI for each habitat type h with h = 1, . . . n; where n = number of different habitats in the considered protected area level (single SAC or the overall NETwork).
- Standardization of the NDVI distribution for each pixel of the given habitat type h within the protected area:

$$\overline{\text{NDVI}}_{\text{SAC}} = \frac{\text{NDVI} - \mu_h(\text{NDVI})}{\sigma_h(\text{NDVI})} \qquad (4)$$

where SAC indicates the statistics estimated at site level; similarly, $\overline{\text{NDVI}}_{\text{NET}}$ is computed on the basis of statistics ($\mu_h$ and $\sigma_h$) at the local Network level. Standardization enables us to directly compare areas where NDVI values have different statistical distributions.

Moreover, in a multitemporal perspective, it allows the comparison of several acquisition dates even in the presence of slight variations of the phenological state.

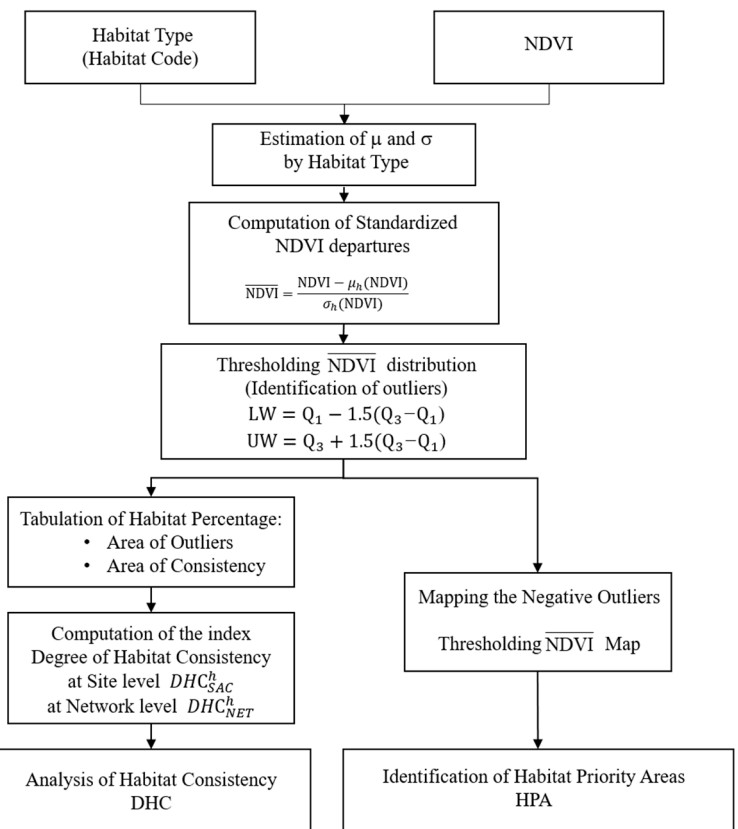

**Figure 2.** Flowchart of the procedure designed for estimating the Degree of Habitat Consistency (DHC) and identifying the Habitat Priority Areas (HPA) at level of a single protected site (SCI/SAC level statistics) or among all the sites of the network (network level statistics).

- Estimation of the LW and UW thresholds for the standardized NDVI distribution ($\overline{\text{NDVI}}$) of each habitat as defined in Equations (2) and (3).
- Mapping the negative outliers to identify the Habitat Priority Areas (HPA) within the SAC by applying the habitat LW thresholds to the $\overline{\text{NDVI}}$ map:

$$\text{HPA}^{h}_{\text{SAC}} \leq \text{LW}^{h}_{\text{SAC}} \tag{5}$$

$$\text{HPA}^{h}_{\text{SAC}} = \sum_{h=1}^{n} \text{HPA}^{h}_{\text{SAC}} \tag{6}$$

similarly, $\text{HPA}_{\text{NET}}$ is computed by thresholding LW derived from the statistics at the local network level.

- Evaluation of the Degree of Habitat Consistency (DHC) for each habitat h with respect to the protected site (SAC) or the network (NET).

$$\text{DHC}^{h}_{\text{SAC}} = \frac{A^{h}_{\text{tot(SAC)}} - A^{h}_{\text{out(SAC)}}}{A^{h}_{\text{tot(SAC)}}} \tag{7}$$

$$\text{DHC}^{h}_{\text{NET}} = \frac{A^{h}_{\text{tot(NET)}} - A^{h}_{\text{out(NET)}}}{A^{h}_{\text{tot(NET)}}} \tag{8}$$

where $A^h_{tot(.)}$ is the total area of the considered habitat at the SAC or at the network level and $A^h_{out(.)}$ is the area where outliers are found.

- Assessment of the global status of the protected areas by considering all the enclosed habitats

$$\text{DHC}_{\text{SAC}} = \sum_{h=1}^{n} \text{DHC}^h_{\text{SAC}} \qquad (9)$$

similarly, $\text{DHC}_{\text{NET}}$ is obtained by summing the area of consistency of all the habitat at the local Network level.

Multitemporal analysis of habitat conditions is based on the changes in HPA and DHC values, the interpretation of habitat modification is also supported by the difference in $\overline{\text{NDVI}}$. For this validation phase, 1985 and 2009 data were compared.

Spatial analysis and other calculations were carried out in the QGIS 3.14.15, PCI Geomatics 9.1 and GRASS 7.8.3 environments. The geographic information was projected to a common UTM zone 33 under the WGS84 datum.

2.3.2. Validation Procedure

The first step for validating the results of the proposed procedure was based on a careful check of reports from field surveys. The type and severity of degradation phenomena and the presence of natural and anthropogenic stressors, as categorized by field inspection, were spatially associated with the indicated toponyms. By comparing such information with the HPA index and the $\overline{\text{NDVI}}$ maps, a preliminary qualitative assessment of the implemented approach was obtained.

To provide a quantitative evaluation of the effectiveness of the proposed procedure, information from field surveys was integrated in a map based on orthophotos, where areas affected by evident degradation phenomena (fragmented or sparse vegetation, burnt and barren areas, etc.) were localized by visual interpretation and digitalized.

The digitized vector map was then rasterized to overlap with the Landsat-based map of anomalous areas. Errors inherently associated with the digitization process can be considered negligible due to the higher spatial resolution of the orthophotos (less than one meter) in comparison with that of the 30 m Landsat pixel [67].

Ultimately, the digitized map including areas depicted as degraded by field reports was compared with the map showing those areas identified as potentially anomalous by the procedure (taking into account only negative outliers). From the comparison, the error matrix was derived, and then the overall accuracy (A) as well as omission (Producer's Accuracy) and commission (User's Accuracy) errors were estimated (see, e.g., [68,69]).

## 3. Results

### 3.1. Accuracy and Exportability

The estimation of the accuracy of the proposed procedure was performed using the extensive in situ observations (data, maps, photographs, reports, etc.) compiled in the first phase of the regional monitoring plan of the Natura 2000 network of Basilicata together with the 1988 and 2008 aerial photographs (1:10,000) with less than 1 m of spatial resolution.

The comparison of the negative anomalies identified by satellite and the ground truth derived by integrating orthophoto digitalization and field reports showed that most of pixels identified as anomalous ($\approx$84% of overall accuracy) fall within areas recognized as degraded. About the 13% of anomalies fall out of the validation area (commission errors); such false alarms are mainly linked to those pixels located on the interface between two different habitats (see example in Figure 3). They emerged as anomalous especially if the involved habitats are very different *from* the eco-physiological point of view (e.g., an herbaceous cover adjacent to a forest species). About 7% of non-anomalous pixels were located within scarcely vegetated or evidently degraded areas (omission errors). Such undetected anomalies mainly correspond to small-sized degraded areas (e.g., open-air

micro-dumps, illegal cutting of few plants, small and abandoned quarries, etc.) surrounded by the vegetation of the corresponding habitat. Within the analyzed pixel, the degraded sub-pixel area is not sufficient to produce a macroscopic biomass anomaly at Landsat scale (30 m) and therefore they cannot be detected at such a scale.

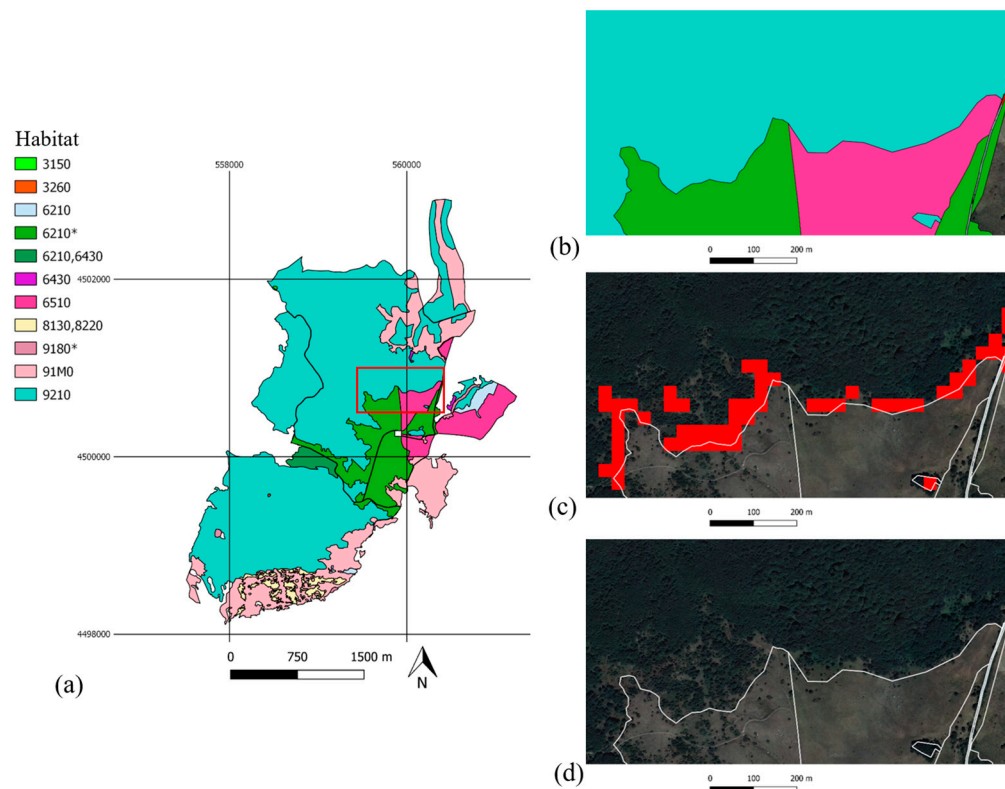

**Figure 3.** Contrasting features of adjacent habitats within the Monti Li Foi SAC: (**a**) Map of habitats; (**b**) zoom on the analyzed habitats (red box): 9210 (*Apennine beech forests with Taxus and Ilex*), characterized by wooded vegetation, borders on herbaceous species 6510 (*Lowland hay meadows—Alopecurus pratensis, Sanguisorba officinalis*); (**c**) map of the 2009 anomalies overlapped with the 2008 orthophoto: pixels belonging to the beech forest and falling at the border between the two habitats are almost all marked as degraded, (**d**) 2008 orthophoto without overlap.

To fill the gap represented by interface pixels, accuracy statistics were recomputed by considering a buffer of one pixel at the boundary between two different categories of habitats. By identifying and masking these pixels, the obtained overall accuracy was considerably higher (~97%) than the previous calculations with reduced commission (about 8%) and omission (about 6%) errors (see Table 2). For the single site, the worst conditions were found in the Bosco Pantano di Policoro SAC (~93%) containing portions of the habitats with a very elongated structure.

**Table 2.** Accuracy metrics for the investigated SACs and for the overall habitats of the three sites.

| SAC | Code | OA | CE | OE |
|---|---|---|---|---|
| Monte Vulture | IT9210210 | 98 | 6 | 5 |
| Monti Li Foi | IT9210215 | 97 | 7 | 6 |
| Bosco Pantano di Policoro | IT9220055 | 93 | 11 | 6 |
| **Overall habitats** | | **97** | **8** | **6** |

These outcomes can be considered satisfactory considering that the proposed approach is conceived as a preliminary screening aimed at intercepting macro-effects of degradation. Areas with such features have to be considered as intervention priorities within the network

where it is advisable to focus dedicated field campaigns to verify the implementation of the adopted conservation measures or to design additional interventions.

The proposed procedure can be easily adjusted to other satellite/airborne sensors by selecting the most appropriate product for the target study in terms of spatial and spectral resolutions. The possible use of the freely available Sentinel-2 data (10 m of spatial resolution) or Very High Resolution (VHR) images at even higher spatial resolutions can notably reduce both commission errors for mixed pixels and the missing detection of small anomalous areas in healthy vegetated environments (omission errors).

### 3.2. Test Sites Habitat Status

To better understand the presence of the identified anomalies, the results were analyzed jointly with reports of field surveys and ancillary data for some SACs representing paradigmatic cases of rich biodiversity of the Natura 2000 ecological network of Basilicata.

For each of the three selected sites (see Section 3.1), the habitat spatial distribution, the map of negative anomalies for the year 2009, and the $DHC_{SAC}^h$ index computed at SAC level are shown.

#### 3.2.1. Monte Vulture

The map of the anomalies for 2009 shows many zones in different habitats involved in possible degradation processes (Figure 4a,b), even though high $DHC_{SAC}^h$ values have been obtained throughout the site (always above 90% apart from the habitat 91B0—*Thermophilous Fraxinus angustifolia woods*, magenta in Figure 4a), demonstrating the general good conditions of this site (Figure 4c).

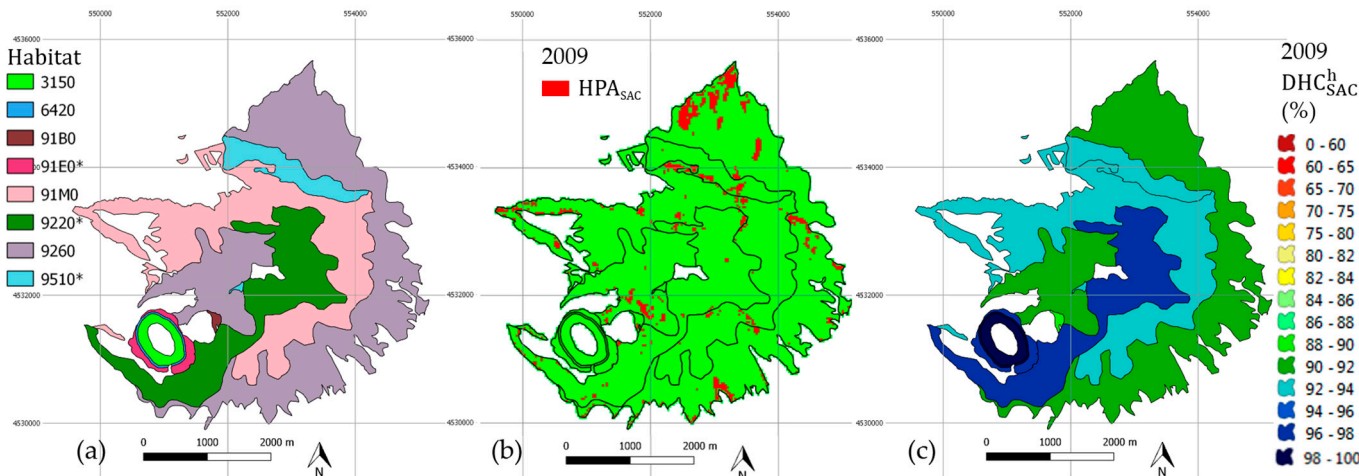

**Figure 4.** Monte Vulture SAC: (**a**) Map of habitats; (**b**) map of habitat priority areas; (**c**) map of the degree of habitat consistency (DHC) index. Maps (**b**,**c**) are obtained on the statistics of 2009 data at SAC level.

As for anomalous areas, the habitats 9260 (*Castanea sativa woods*, light purple in Figure 4a) and 91M0 (*Pannonian-Balkanic turkey oak-sessile oak forests*, pink in Figure 4a) which cover most of the Monte Vulture SAC, are characterized by $DHC_{SAC}^h$ values exceeding 90% but exhibit evident local cases of vegetation stress.

Figure 5a illustrates further details about the areas classified as outliers on the basis of the statistics of the standardized NDVI. In the case of the habitat 9260 (Figure 5b), the identified vegetation anomalies can be ascribed to the improper management of coppice areas with poor topsoils that have been left abandoned in the past. These areas appear as transitory stands towards high forest with the presence of many deteriorated or died trees (for further details, see Supplementary Materials Section S2.1). In the case of the habitat 91M0, the main degradation cause is the management of grazing activities. More precisely,

the high intensity of cattle grazing causes long-term defoliation of native (and protected) plant communities, also facilitating the presence of invasive alien species (e.g., *Asphodelus Microcarpus* and *Asphodelus Albus*) found throughout this habitat (see the monitoring phase report of the Ecological Network of Basilicata at: http://www.retecologicabasilicata.it/ambiente/site/portal/detail.jsp?sec=107281&otype=1012&id=109488, last accessed on 18 March 2022).

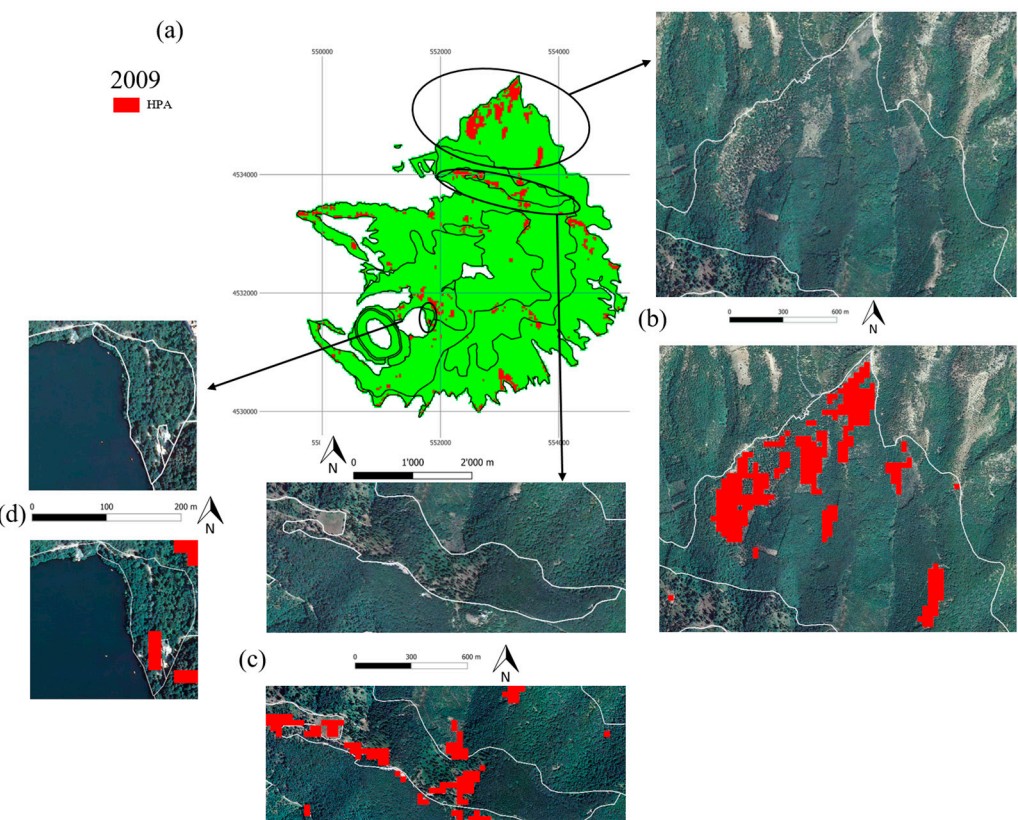

**Figure 5.** Monte Vulture SAC: (**a**) Maps of relative anomalies (2009); zoom of negative anomalies superimposed on the orthophoto within (**b**) habitat 9260 (*Castanea sativa woods*) due to abandonment of marginal areas and bad management of forest areas; (**c**) habitat 9510* (*Southern Apennine Abies alba forests*) due to grazing pressure and unfavorable biotope conditions; (**d**) habitat 91B0 (*Thermophilous Fraxinus angustifolia* woods) due to tourism flows favoring the growth of synanthropic and alien species.

As for habitat 9510* (*Southern Apennine Abies alba*, cyan in Figure 4a), two different patches can be distinguished. The first one is a very small area located at the center of the site. Here, trees grow under good conditions (Figure 5a), whereas the second one is vast and located on the northern slope of the Monte Vulture (Figure 5c). On the contrary, these populations are in a dreadful state, showing several cores of stressed plants. These specimens, probably planted at the beginning of 20th century due to the Zanardelli Law (to protect mountainous areas from hydrogeological instability [70]), suffer from both the effects of unsustainable grazing pressure and unfavorable conditions of the biotope in terms of soil and humidity.

The comparison between figures puts into evidence those areas marked as negative outliers that are characterized by sparse or fragmented vegetation cover.

Tourism escalation also creates increasing disturbance to the habitat, especially in the vicinity of the lakes, where many activities devoted to accommodation and food services are located. In this context, the negative anomalies detected within the limited extent of the habitat 91B0 (*Thermophilous Fraxinus angustifolia* woods, brown in Figure 4a) are possibly the effects of hit-and-run tourists that favor the occurrence of several synanthropic and

alien species, limiting the naturalness and the well-being of the habitat (see Figure 5d, [71]). Some signs of clustering near the main roads seem to confirm the existence of anthropic pressure behind the critical areas.

### 3.2.2. Monti Li Foi

The general conditions of the habitats of the Monti Li Foi SAC (Figure 6a) are satisfactory even though some areas deserve to be carefully monitored (Figure 6b,c).

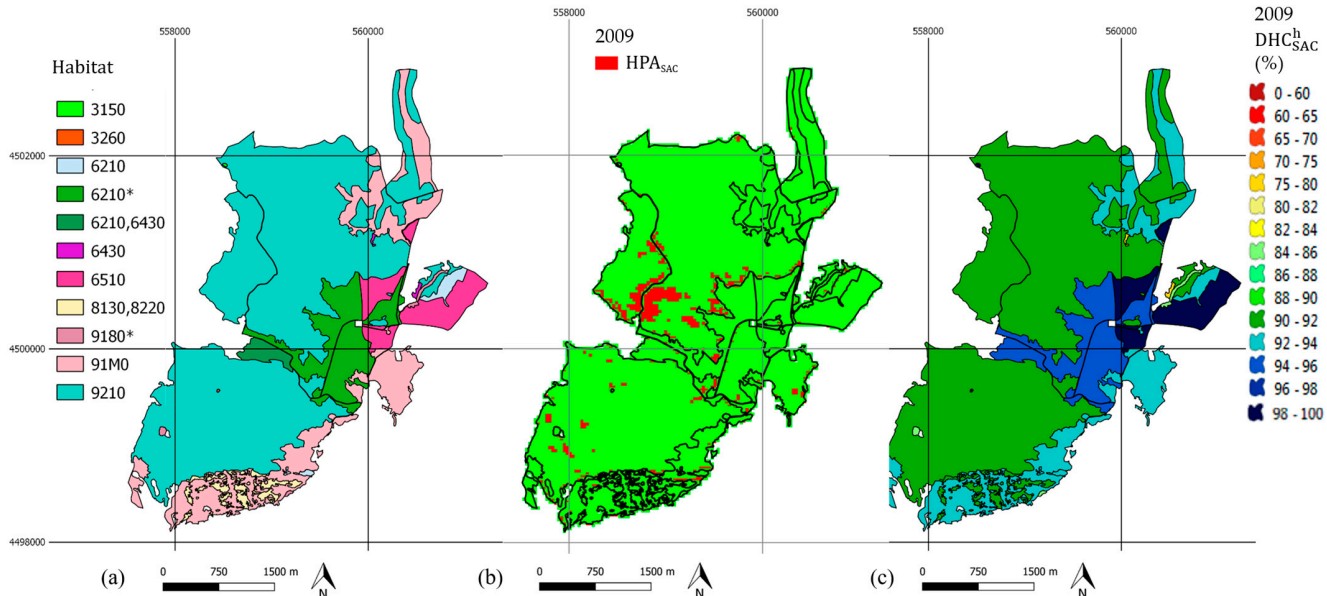

**Figure 6.** Monti Li Foi SAC: (**a**) Map of habitats; (**b**) map of habitat priority areas; (**c**) map of the degree of habitat consistency (DHC) index. Maps (**b**,**c**) are obtained on the statistics of 2009 data at SAC level.

Especially for the dominant habitat 9210 (*Apennine beech forests with Taxus and Ilex*, dark cyan in Figure 6a) many areas are characterized by forest expansion due to colonization of herbaceous covers within and at borders of the forests, as observed in other forest stands of Basilicata [72]. In spite of this situation, a core of negative anomalies persists in the central part of the SAC (Figure 7a) even if the presence of critical areas is rather circumscribed. In fact, most of these formations are relict forests, composed of populations reduced in number and size and recognized as in need of special conservation measures [73]. In these areas, the long-term lack of silvicultural practices in combination with the detrimental effects of grazing (for this site, mainly observed in forest stands; for further details, see Supplementary Materials) are the potential driving forces inducing vegetation decline (in Figure 7c, most of the detected anomalies correspond to sparsely vegetated covers; see the monitoring phase report of the Ecological Network of Basilicata at: http://www.retecologicabasilicata.it/ambiente/files/docs/DOCUMENT_FILE_102156.pdf, last access on 18 March 2022).

On the opposite, notable $DHC_{SAC}^h$ values (over 95%, Figure 7b) are reached by the habitats 6210* and 6510 (*semi-natural dry grasslands and scrubland facies on calcareous substrates—Festuco-Brometalia* with important orchid sites and *lowland hay meadows—Alopecurus* pratensis, *Sanguisorba officinalis*, respectively green and fuchsia in Figure 7a). The former holds important orchid communities of considerable floristic and phytogeographic value. The latter are man-managed grasslands located at high elevations (about 1200–1300 m a.s.l.), and their considerable ecological value is due to their exceptional occurrence within the landscapes of Basilicata. In particular, this habitat represents an exemplary case of how human practices can be essential for the maintenance of some typical ecosystems. The good state of preservation of these grassland formations is guaranteed just by the execution of farming practices, including mechanical mowing and sometimes fertilization [74]. The abandon-

ment of this kind of practice or, on the contrary, increasing levels of exploitation would imply the decline of species richness and the transformation of this valuable habitat into a more ordinary mesophile/hygrophilous grassland [75,76]. Additionally, in the case of the Monti Li Foi SAC, some clusters located in the proximity of roads suggest the presence of anthropogenic stress on the natural habitat.

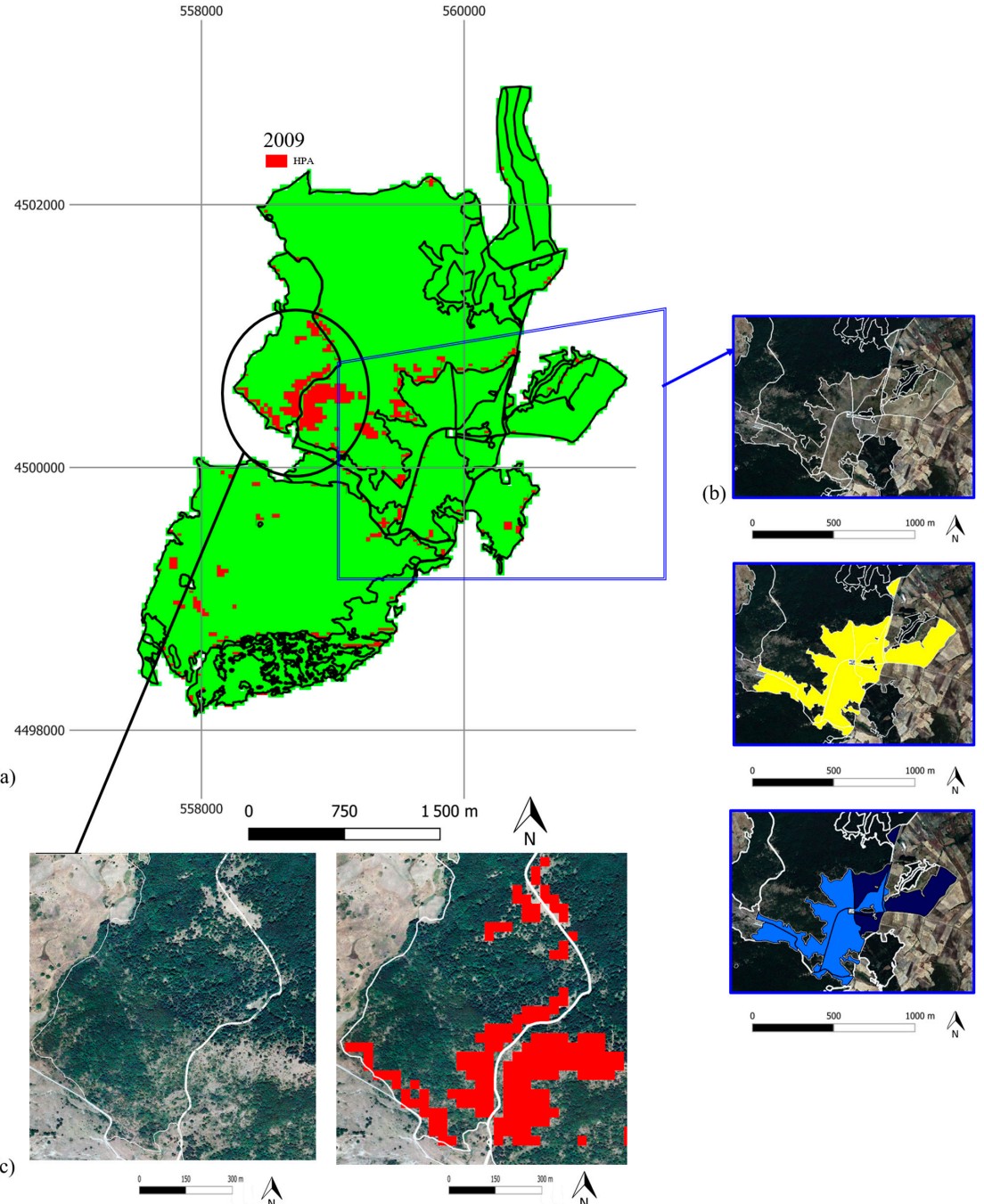

**Figure 7.** Monti Li Foi SAC: (**a**) Map of relative anomalies (2009); zoom of negative anomalies superimposed on the orthophoto within (**b**) habitat 6210* and 6510 (*Semi-natural dry grasslands and scrubland facies on calcareous substrates—Festuco-Brometalia with important orchid sites and Lowland hay meadows—Alopecurus pratensis, Sanguisorba officinalis*), highlighted in yellow, as examples of good vegetation conditions maintained, in the case of the habitat 6510, by sustainable agricultural practices; (**c**) habitat 9210 (*Apennine beech forests with Taxus and Ilex*) due to overgrazing along with the lack of silvicultural practices.

### 3.2.3. Bosco Pantano di Policoro

The Bosco Pantano di Policoro SAC (Figure 8a) shows limited disturbance due to tourism and recreational activities (see the pattern of 2009 anomalies (HPA) and the $\mathrm{DHH}_{\mathrm{SAC}}^{h}$ map in Figure 8b,c); however, it has undergone a massive land reclamation since the Thirties. This consisted of several extensive canalizations aimed at gaining lands for anthropic activities (mainly agriculture, [77]). Moreover, since the 1960s, urbanization and overexploitation of groundwater for agricultural purposes have accelerated degradation phenomena such as salinization, coastal erosion, loss of soil productivity, and the depletion of natural resources [78–80]. This was only partially counterbalanced by the approval of specific legislations for protecting environmentally valuable areas (e.g., National Law 431/1985 for the protection of specific landscapes and surroundings areas; EU Water Framework Directive 2000/60—Integrated River Basin Management for Europe).

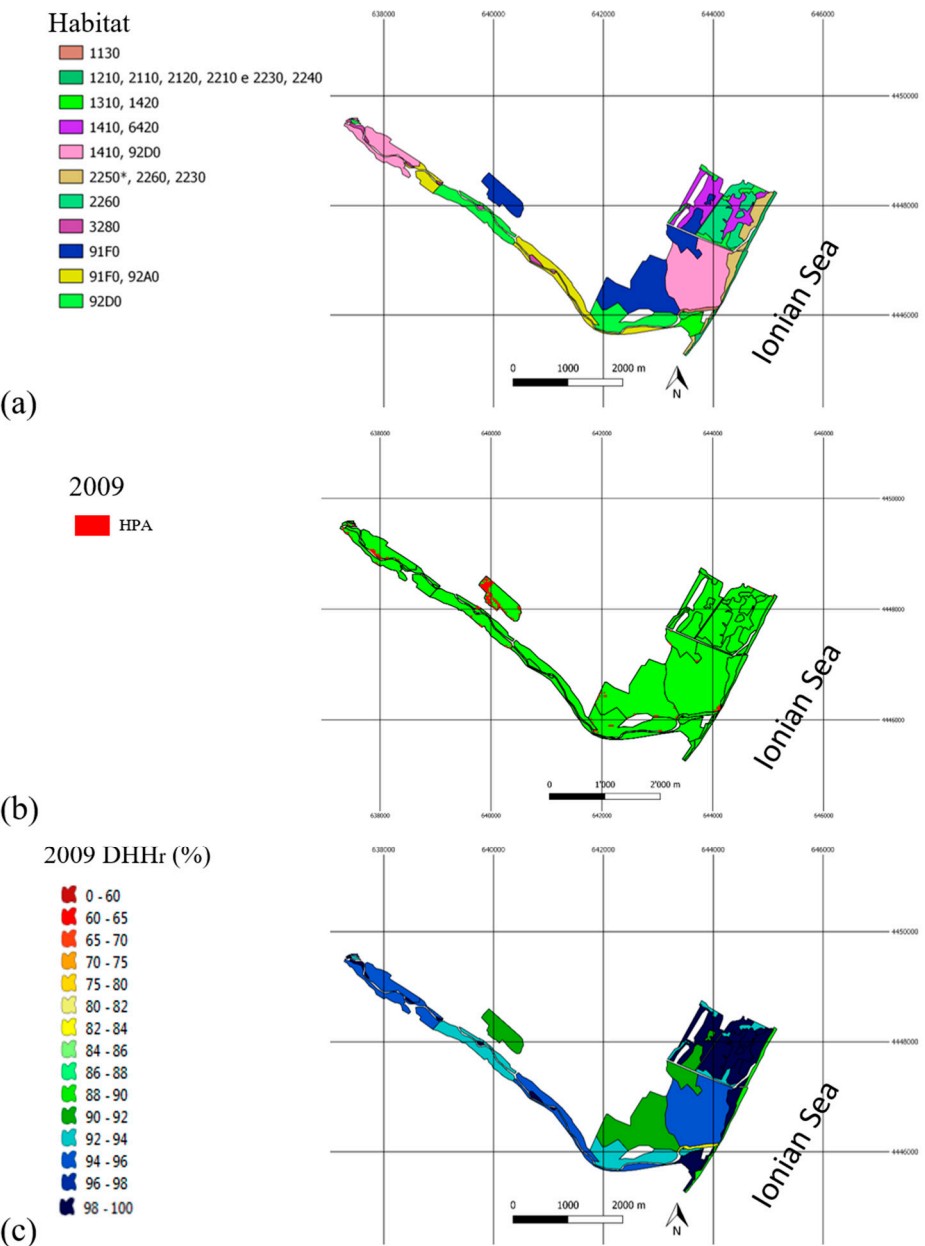

**Figure 8.** Bosco Pantano di Policoro SAC: (**a**) Map of habitats; (**b**) map of habitat priority areas; (**c**) map of the degree of habitat consistency (DHC) index. Maps (**b**,**c**) are obtained on the statistics of 2009 data at SAC level.

Some scars of past human interventions are still visible in the area, especially on riparian vegetation. In greater detail, the habitat 91F0 (*Riparian mixed forests of Quercus robur, Ulmus laevis* and *Ulmus minor, Fraxinus excelsior* or *Fraxinus angustifolia*, blue in Figure 8a), 92D0 with 1410 (*southern riparian galleries and thickets with Mediterranean salt meadows—Juncetalia maritime*, pink in Figure 8a), and 92A0 (*Salix alba and Populus alba galleries*, dark yellow in Figure 8a) often bordering each other, represent the last remnants of one of the most important planitial woods of Southern Italy, containing a range of ecological and biological features similar to those that can be found in tropical environments [81].

The vegetation patch distribution (Figure 9a–c) is strongly influenced by the presence of marshy areas as a key factor for the growth of riverine vegetation. In the 2009 image there are few permanent flooded areas in what was once a significant and stable wetland (see the monitoring phase report of the Ecological Network of Basilicata at: https://fdocumenti.com/reader/full/secondo-report-luglio-settembre-2009-area-8-b-plantago-lagopus-lagurus-ovatus, last accessed on 30 March 2022). Hydrological oscillations, derived directly (land reclamation interventions) and indirectly (building of the Monte Cotugno dam and other water regulation works on the Sinni River) from anthropic actions, result in a reduced river flow [82]. Consequently, although vegetation has resumed growth with respect to previous years, some areas appear partially fragmented due to salinization phenomena and the consequent presence of several halophytic species that are considered detrimental to the native hygrophilous woods [83].

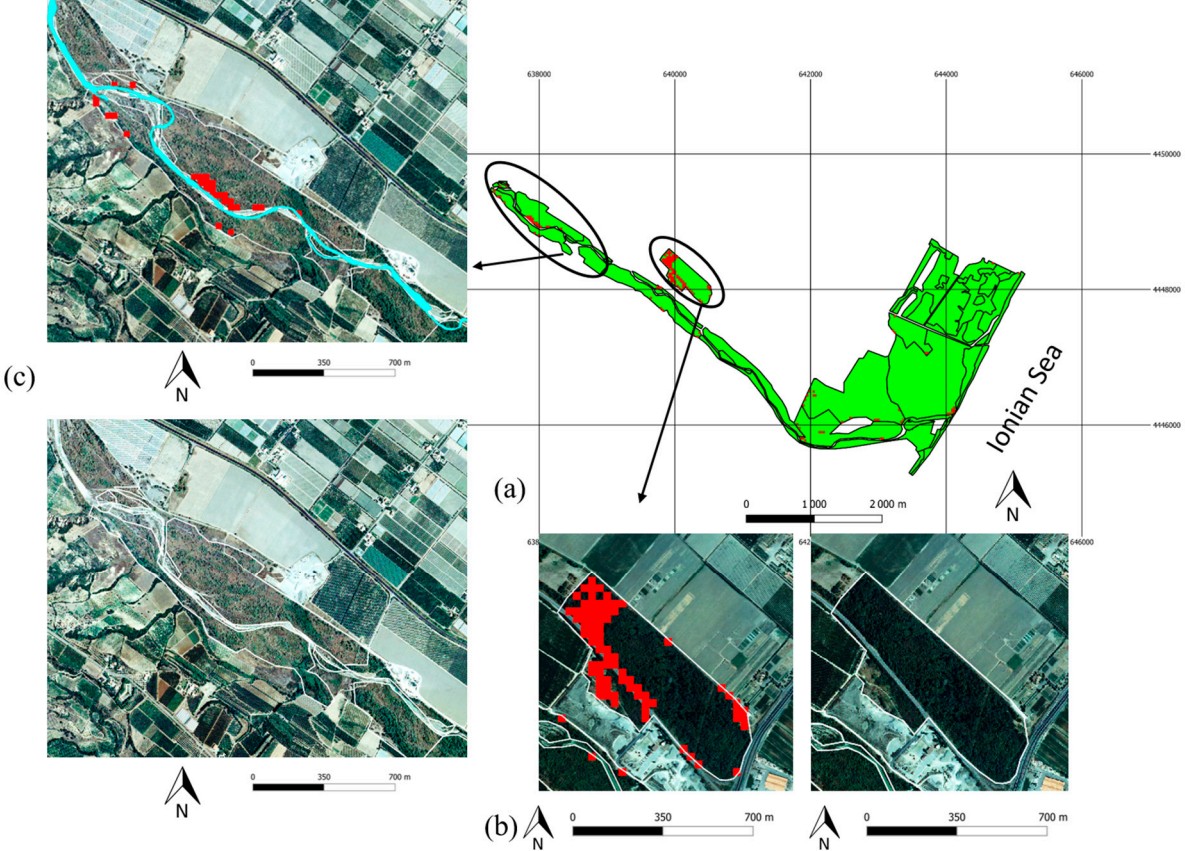

**Figure 9.** Bosco Pantano di Policoro SAC: (**a**) Map of relative anomalies (2009); zoom of negative anomalies superimposed on the orthophoto within (**b**) habitat 91F0 (*Riparian mixed forests of Quercus robur. Ulmus laevis and Ulmus minor, Fraxinus excelsior, or Fraxinus angustifolia, along the great rivers*), associated with the fragmented structure of vegetation patches due to salinization phenomena and invasion of halophytic species; (**c**) habitat 92D0 with 1410 (*southern riparian galleries and thickets with Mediterranean salt meadows—Juncetalia maritime*) as a consequence of hydrological oscillations in water regime (the Sinni River, in cyan, crosses the habitat).

### 3.3. General Conditions of the Overall Natura 2000 Network of Basilicata

According to the methodology described in Section 2.3.1, the assessment of Habitats Consistency (DHC) and the identification of priority areas (HPA), i.e., areas showing signs of degradation and needing particular attention for in situ investigations and interventions, is based on the statistics of the standardized parameter $\overline{\text{NDVI}}$ estimated at network level. Figure 10 shows an example of the standardized NDVI based on these statistics. By thresholding this map and tabulating the surface of the areas of consistency ($\text{DHC}_{\text{NET}}$) and the areas of outliers, the general habitat conditions of the network were assessed (Table 3). The map of priory areas ($\text{HPA}_{\text{NET}}$) of 2009 based on negative outliers can be downloaded in geotif format as Supplementary Material (Map S1).

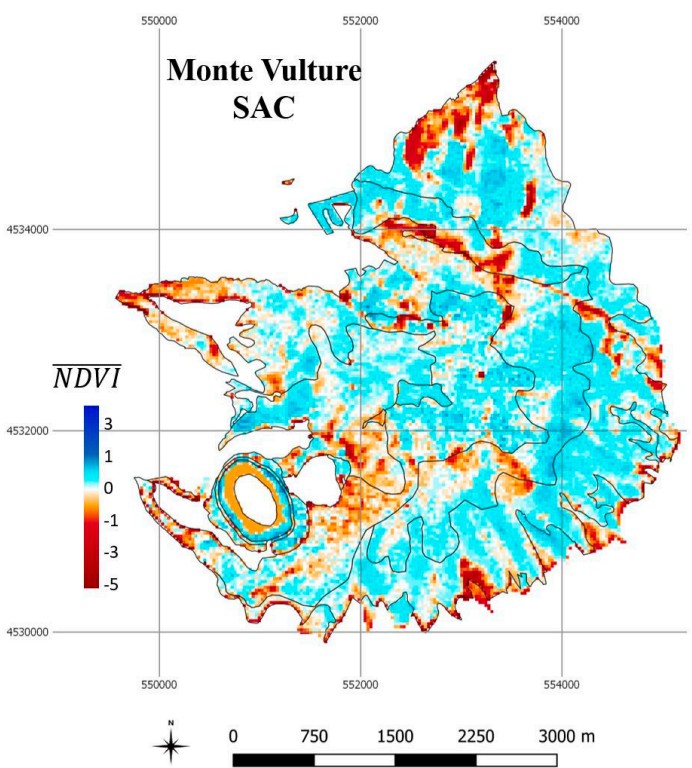

**Figure 10.** Example of a map of the standardized NDVI ($\overline{\text{NDVI}}$) for the SAC of Monte Vulture. Elaboration based on the Landsat image acquired on 27 July 2009.

**Table 3.** Degree of habitat consistency at network level ($\text{DHC}_{\text{NET}}$) and anomalous (positive and negative) areas for 2009 and 1985 for the investigated Natura 2000 SAC of Basilicata region.

|  | 2009 (%) | Extent 2009 (ha) | 1985 (%) | Extent 1985 (ha) |
|---|---|---|---|---|
| Negative Outliers | 5.36 | 608.14 | 3.78 | 428.80 |
| $\text{DHC}_{\text{NET}}$ | 93.78 | 10,637.88 | 95.69 | 10,854.66 |
| Positive Outliers | 0.86 | 97.56 | 0.53 | 60.12 |
| Total | 100 | 11,343.58 | 100 | 11,343.58 |

For the year 2009 (Table 3), the totality of the habitats present in the network reached a level of consistency of about 94% ($\text{DHC}_{\text{NET}}$); this means that a very high percentage of the protected surfaces is occupied by habitats with a vegetation matrix that conforms to the average habitat value. The percentage of negative outliers (about 5.4%) indicates that about 600 hectares of the total protected surfaces are occupied by habitat portions where the vegetation matrix has a structure and vigor below the mean conditions. Thus, it suggests that degradation processes are acting in these areas and detailed in situ investigations are required.

Data for the year 1985 show a distribution quite similar to 2009 (Table 3), but with a level of consistency of the habitats slightly higher (DHC$_{NET}$ ≈96%) and a lower presence of critical areas (about 180 hectares less). Therefore, during the period 1985–2009, i.e., before the identification of protected sites and after the confirmation of the SCIs, a worsening of habitat conditions at network level is observable. Generally, the patches of critical areas are the same for the two years, but they are wider in 2009; such behavior was also found at SAC level as described in the following sections.

By analyzing the detail of the involved habitats, seven habitats cover about 50% of these priority areas (Table 4). In particular, the forest habitats (9260, 91M0 and 9220*), being the largest ones, include most of these priority areas, whereas the herbaceous habitats 6430 (*hydrophilic marginal communities of high plains and from mountain to alpine*) have the highest percentage of critical areas (12.5%) with respect to their dimension.

**Table 4.** Habitats showing the highest percentage of negative anomalous areas (habitat priority areas) computed at network level for the year 2009 and 1985.

| Habitat Code | Description | Anom. Area 2009 (%) | Anom. Area 2009 (ha) | Anom. Area 1985 (%) | Anom. Area 1985 (ha) | N. of Patches |
|---|---|---|---|---|---|---|
| 6430 | *Hydrophilous tall herb fringe communities of plains and of the montane to alpine levels* | 12.5 | 0.165 | 0 | 0 | 4 |
| 9260 | *Castanea sativa woods* | 9.23 | 63.94 | 7.34 | 50.84 | 3 |
| 91M0 | *Pannonian-Balkanic turkey oak–sessile oak forests* | 8.78 | 114.04 | 8.34 | 108.35 | 172 |
| 91F0 | *Riparian mixed forests of Quercus robur, Ulmus laevis, and Ulmus minor, Fraxinus excelsior, or Fraxinus angustifolia, along the great rivers* | 7.64 | 11.01 | 7.32 | 10.55 | 6 |
| 9220* | *Apennine beech forests with Abies alba and beech forests with Abies nebrodensis* | 7.61 | 89.58 | 6.23 | 73.34 | 443 |
| 3290 | *Intermittently flowing Mediterranean rivers of the Paspalo-Agrostidion* | 7.58 | 1.41 | 3.32 | 0.61 | 7 |
| 9510* | *Southern Apennine Abies alba* | 7.49 | 5.53 | 0.86 | 0.64 | 2 |
| 9210 | *Apennine beech forests with Taxus and Ilex* | 6,93 | 36.28 | 8.9 | 46.6 | 35 |
| 91E0* | *Alluvial forests with Alnus glutinosa and Fraxinus excelsior* | 2.6 | 0.57 | 9.52 | 2.09 | 3 |

## 4. Discussion

Specific results obtained in the investigated Basilicata Natura 2002 network confirm the relevance of adopting specific conservation measures; a simple delineation of a protected area is not enough [84]. The multi-temporal analyses at network level highlighted a worsening of habitat conditions before the identification of protected sites and after the confirmation of the SICs (i.e., before the adoption of conservation measures plans, and thus the conversion in SAC). Moreover, results at site level support the need for continuous management of protected sites since peculiar species can be further threatened with a passive conservation approach [85], such as in the case of forest habitats in Monte vulture SAC (9260—Castanea sativa woods) and in Monti Li Foi SAC (9210—Apennine beech forests with Taxus and Ilex).

To preserve a good status of habitat conditions, nowadays, controlled disturbance regimes are being discussed in forest ecosystems [85], but they are more crucial for transitional habitats [86]. In conservation areas, lower-intensity management such as reduction or exclusion of grazing, or decreased shrub clearing can promote a gradual conversion of herbaceous to shrub and woody species. Such encroachment processes can rapidly deteriorate habitats, altering ecosystems' function, especially tree–grass coexistence, and threatening grassland-related species [87]. A positive implementation of controlled disturbances was found in Monti Li Foi grasslands (habitat 6510), where extensive farming practices maintain an equilibrium in ecosystem diversity and functionality, avoiding encroachment processes largely diffused in many mountain areas due to land abandonment [63,72].

From a methodological point of view, in light of the results obtained, the development strategy adopted to implement the procedure confirms the relevance of collaborations

between ecologists, land managers, and remote sensing experts to identify and define tailored procedures for operatively adopting remote-sensing satellite data in ecological monitoring programs [8,17,88]. Such cooperation is relevant both in the design (tailoring ecological requirements with remote-sensing data spatial and temporal characteristics) and validation phases (ecological interpretation of remote sensing findings) [5,88].

From a technical point of view, the implemented procedure proved to be able to identify critical areas related to both drastic changes and incipient modifications inside the habitat with homogeneous (e.g., habitat 9260) and patchy (e.g., habitats 6420, 6210*) cover-type. Such a capability largely encompasses results of the generally used approach based on land-cover changes on broad habitat types (forest, grassland, shrub, etc.) being able to identify only modifications that imply a complete change of cover [22,89]. Pixel-based trend analysis based on Sentinel-2 can instead provide similar results; tools such as NaturSat [13] are very interesting for capturing the peculiar structure and complexity of protected sites.

The export of the proposed smart procedure to Sentinel-2 data with their 10 m spatial resolution can improve results (reduced commission errors, see Section 3.1) in habitats with very elongated and narrow shape, such as in riparian areas [15]. Obviously, they are not able to support historical data analysis as Landsat or SPOT to assess the baseline conditions of Natura 2000 establishment. Tools for Landsat such as LandTrendr (also available in GEE—Google Earth Engine [90]) can be biased by the discrete temporal sampling of Landsat data (16 days) inducing little shifts in phenology caused by different acquisition dates of cloud-free imagery or by the peculiar meteo-climatic conditions of a given year. Hyperspectral satellite data (e.g., PRISMA [91]) that suffer from the same discrete temporal sampling at the nadir and heavy processing time can benefit from a single date procedure as the one proposed

In the last few years, alongside the direct use of remote data to provide a picture of the health status of protected areas, another research line has begun to receive more and more attention: the estimation of the habitat quality by mapping and quantifying (also in monetary values) ecosystem services provided by natural areas (e.g., through InVest software [92,93]). These works are generally based on land-cover data, but an attempt to integrate estimates on habitat consistency and/or intra-habitat variability of vegetation cover status from satellite data could be promising.

## 5. Conclusions

The development of monitoring strategies and the implementation of specific policies to preserve biodiversity in a changing world are crucial tasks because of the increasing impact of anthropogenic activities and climate change on natural resources.

In the present work, the assessment of habitat conditions was carried out by testing the reliability and efficacy of a straightforward and cost-effective index (DHC) based on satellite-derived vegetation index and GIS techniques. The DHC focuses on the statistical distributions of a given habitat (at the site or network level) and, considering the state of consistency of all the habitats present, it provides a value that summarizes the general status of the protected site and of the entire network. In this way, it is possible to compare environmental conditions of a certain habitat within a given protected site (SCI, SAC) with those belonging to the other sites of the network.

At the same time, the identification of the spatial distribution and extent of degraded areas in the habitat vegetation matrix (Habitat Priority Areas) represents a useful support for protected area managers and policy makers. This information layer indicates the areas in which to concentrate in-depth field investigations and expedites the evaluation of the adopted conservation measures. As an example, for the investigated Natura 2000 network of Basilicata, most of the identified critical areas concern mainly the effects of grazing, poor forest-management practices, past hydraulic interventions (construction of dams and small river barrages), and conservation activities (reforestation plans with non-native species, land reclamation works, etc.).

The good performances of the proposed procedure (overall accuracy $\approx$ 83% rising to $\approx$97% with one pixel mask at the boundary between habitats) suggest its use in the operational monitoring of protected areas both in the starting phase (mainly focused on the characterization of habitats) to rationalize field-survey resources, and in the subsequent assessment phase (periodic reporting, as established by the EU directive) to evaluate the effectiveness of conservation measures. Its structure allows an application independent of specific/local conditions, and it can be easily exported to other protected areas and adapted for applications with sensors that have different (and higher) spatial/spectral features such as the multispectral Sentinel 2 data with 10 m of spatial resolution for the NIR and RED bands or the hyperspectral PRISMA data at 30 m of spatial resolution.

The DHC can represent a comprehensive index of habitat status that can be promptly added to the EU standard form to evaluate the conservation effectiveness of the EU initiatives in the perspective to conceive the safeguard of biodiversity as a great opportunity for socioeconomic and cultural growth.

**Supplementary Materials:** The following supporting information can be downloaded at: https://www.mdpi.com/article/10.3390/rs14112699/s1, Map S1: GeoTif of Habitat Priority Areas (HPA) at network level (GeoTif format, year 2009); Section S2: Multitemporal analysis of habitat status within test sites. Citations of Section S2 are included in the Reference section [94–97].

**Author Contributions:** Conceptualization: T.S. and V.I.; methodology: T.S., M.L. and V.I.; collection, selection, and analysis of satellite data: T.S., R.C. and V.I.; supervision: T.S.; writing—original draft preparation, V.I., T.S. and M.L.; writing—review and editing: T.S., V.I., R.C. and M.L.; funding acquisition, T.S. and V.I. All authors have read and agreed to the published version of the manuscript.

**Funding:** This research was partially funded by the project "Basilicata Natura 2000 Network" within the "Cabina di Regia Natura 2000" approved by the Ministry of the Environment and Protection of Land and Sea of Italy (Ministerial Decree 224/202) and financed under the Measures 1.4 of the Basilicata Regional Operational Programme (ROP) 2000–2006, the 2014–2020 Rural Development Programme of Basilicata Region (Misura 16.1, InnForestGo, CUP C36C18000420006 and For-E.So.Carb, CUP F44E19000280002).

**Data Availability Statement:** The data presented in this study are available on request from the corresponding author.

**Acknowledgments:** We would thank the Biodiversity and Nature Protection Office of Region Basilicata for providing the reports of field surveys and protected area boundaries. We give special thanks to Antonella Logiurato, referent person for the management of the implementation activities of the Basilicata Natura 2000 Network.

**Conflicts of Interest:** The authors declare no conflict of interest. The funders had no role in the design of the study; in the collection, analyses, or interpretation of data; in the writing of the manuscript, or in the decision to publish the results.

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
