# Peer review of "A Smart Procedure for Assessing the Health Status of Terrestrial Habitats in Protected Areas: The Case of the Natura 2000 Ecological Network in Basilicata (Southern Italy)"

_remotesensing, doi:10.3390/rs14112699_

Round 1
Reviewer 1 Report
The paper entitled "A smart procedure for assessing the health status of terrestrial habitats in protected areas: the case of Natura 2000 ecological network in Basilicata (Southern Italy)" is about the analysis of general quality conditions of terrestrial habitats using a methodology (a smart procedure) based on two remote sensing images in hilly, mountainous and coastal plain environments. The results of the spatially explicit assessment are the incoming degradation and the overall habitat status. The paper is of interest for "Remote Sensing" journal but I have some comments and doubts. Concretely, they are:
- An important issue in using Natura 2000 data is the predictable heterogeneity inside each habitat, as recognized by the authors in lines 329, 330 and 331. This heterogeneity can produce misinterpretations. Another can be problem can be the border effect, being visible in figure 13b for instance. Have you considered applying a buffer and a mask? Reading section 5.3. the answer seems yes. So, from my point of view, the "Accuracy" section should be moved to the beginning of the results and all the figures should show the boundary effect out.
- The authors expose an issue in the explanation of the right-tail outliers (lines 329 and 330) because "they cannot be directly related to degradation" but to habitat heterogeneity. Nevertheless, nothing is said of how to solve/minimize it.
- From a methodological point of view I have some doubts if just using some NDVI thresholds you can get similar results. In fact, many works have used this index to assess the environmental impacts, for instance, of forest fires because they produce the most NDVI negative values, clearly contrasted with the rest of pixels.
- The introduction section requires some end points to fluff up the text, especially from lines 40 to 81.
- From lines 106 to 114 of the Introduction section, the authors provide specific information about the applied methodology. This specific text has not sense of being included in that section.
- In figure 1 you can add the Landsat path
- Avoid in the subtitles of sections 3.1.1, 2 and 3 including an alphanumeric code.
- Which software was used to convert digital numbers into radiance and into reflectance?
- In line 312 it appears a text alone, without any sense.
- The original reference of NDVI formula should be added.
- I think that formula 4 corresponds to a process of "standardization" more than normalization. Please, check it out.
- Please, could you include the 1989 results in table 5? The comparison with both years will be clearer. Another improvement would be the use of a colour palette in figures 6b, 6c, and 6d; 8b, and 8c; 10b, and 10c; 12a, and 12b, because they are very dark and nothing is visible.
Author Response
Comments and Suggestions for Authors
The paper entitled "A smart procedure for assessing the health status of terrestrial habitats in protected areas: the case of Natura 2000 ecological network in Basilicata (Southern Italy)" is about the analysis of general quality conditions of terrestrial habitats using a methodology (a smart procedure) based on two remote sensing images in hilly, mountainous and coastal plain environments. The results of the spatially explicit assessment are the incoming degradation and the overall habitat status. The paper is of interest for "Remote Sensing" journal but I have some comments and doubts. Concretely, they are:
1(a). An important issue in using Natura 2000 data is the predictable heterogeneity inside each habitat, as recognized by the authors in lines 329, 330 and 331. This heterogeneity can produce misinterpretations.
2. The authors expose an issue in the explanation of the right-tail outliers (lines 329 and 330) because "they cannot be directly related to degradation" but to habitat heterogeneity. Nevertheless, nothing is said of how to solve/minimize it.
We thank the reviewer for highlighting this point, which allowed us to improve our paper. Natura 2000 habitats show some degrees of heterogeneity that are not expected to impact on the habitat health. Our statistical analysis focuses on photosynthetic anomalies that are statistically inconsistent with the large part of the cover of the investigated area. Pixels identifiable by outliers may emerge from heterogeneity, but they are interesting mainly because of their anomalous photosynthetic activity. If they are on the left tail of the distribution, then the vegetation health/density is worse than that observed in the whole area and can suggest the occurrence of land degradation. If instead they are on the right tail, there is not a problem of low health/density. Rather, they may reveal the presence of invasive alien plants or encroachment processes which can act as disturbance factors. In both the cases, our analysis represents a precious support to finalize specific field campaigns that can solve the problem.
We modified the comment to clarify such a concept at lines 278-281 of the new version of the manuscript.
1(b). Another can be problem can be the border effect, being visible in figure 13b for instance. Have you considered applying a buffer and a mask? Reading section 5.3. the answer seems yes. So, from my point of view, the "Accuracy" section should be moved to the beginning of the results and all the figures should show the boundary effect out.
As for the edge effects, as suggested we moved the "Accuracy" section to the first section of the results and substituted the figures by applying the edge mask.
3. From a methodological point of view I have some doubts if just using some NDVI thresholds you can get similar results. In fact, many works have used this index to assess the environmental impacts, for instance, of forest fires because they produce the most NDVI negative values, clearly contrasted with the rest of pixels.
The reviewer is right, the use of thresholds directly on NDVI can produce similar results for changes due to drastic modifications, such as those due to forest fires.
The proposed method, through the standardized NDVI statistics, allows the localization of both drastic modifications (forest fires, cutting) and incipient changes that are anomalous respect to the general behavior of the habitat (dead or dying plants in the habitat 9260 of the Monte Vulture SAC, see figure S.2.c). Moreover, by accounting for the intra-habitat variability, it allows us to minimize phenological effects due to differences in acquisition time and to identify adaptive thresholds based on the specific habitat characteristics and phenological conditions; mere NDVI thresholds could be phenology dependent and require manual adjustments.
4. The introduction section requires some end points to fluff up the text, especially from lines 40 to 81.
This paragraph has been rewritten.
5. From lines 106 to 114 of the Introduction section, the authors provide specific information about the applied methodology. This specific text has not sense of being included in that section.
Details on the applied methodology have been removed from the introduction.
6. In figure 1 you can add the Landsat path
Information on the Landsat path has been added in figure 1 (which is also rearranged according to reviewers 3 and 4).
7. Avoid in the subtitles of sections 3.1.1, 2 and 3 including an alphanumeric code.
N2k alphanumeric codes have been removed.
8. Which software was used to convert digital numbers into radiance and into reflectance?
We used the EASI modeling tool of PCI Geomatics to manually implement the conversion. Such an information has been added to the text (line 233 of the new manuscript).
9. In line 312 it appears a text alone, without any sense.
The misprint has been eliminated.
10. The original reference of NDVI formula should be added.
The reference Rouse et al.,1973 has been added.
11. I think that formula 4 corresponds to a process of "standardization" more than normalization. Please, check it out.
We thank the reviewer for such a suggestion; the general standardization term is more appropriate for our application. It has been replaced.
12. Please, could you include the 1989 results in table 5? The comparison with both years will be clearer. Another improvement would be the use of a colour palette in figures 6b, 6c, and 6d; 8b, and 8c; 10b, and 10c; 12a, and 12b, because they are very dark and nothing is visible.
Table 5 has been updated with 1985 data.
We apologize for the dark contrast of orthophotos, but it is an effect of pdf conversion. Anyway, we increased the contrast to minimize the effects of conversion.
However, as for the first submission, we have also sent all the figures at full resolution in tif format which should be accessible via the MDPI platform or alternatively via links to be requested from the editor.
Reviewer 2 Report
The article refers to the habitat quality assessment of Natura 2000 network of the Basilicata region.
As a whole, it presents quite an interesting study. Before its publication, I propose to make a few corrections.
- The Abstract was written too broadly. Topics related to the idea of ​​the Natura 2000 should be kept to a minimum. The achieved results should also be described in one/two sentences.
- The Introduction contains the necessary information about the background of the issue. 46 literature items were analyzed in it. There are 98 of them throughout the article. At the end of this chapter, please clearly define the purpose of the work.
- The Methodology has been described in a satisfactory and clear manner.
- I propose to separate the Results from Discussion as the Discussion has not actually been carried out. It should refer to how the presented results compare to other similar studies. It should be written at a relatively high level of generality.
- The article should be written impersonally avoiding the words "we", "our" etc.
- Please check all units, as in some places there is no space between the number and unit (see for example lines 72, 81).
- In References. Why was the title of the journal in item 1 written in capital letters? Please also use abbreviations in relation to the names of journals, in accordance with the editorial recommendations.
Author Response
Reviewer 2
The article refers to the habitat quality assessment of Natura 2000 network of the Basilicata region.
As a whole, it presents quite an interesting study. Before its publication, I propose to make a few corrections.
We thank the reviewer for providing suggestions that allowed us to improve our paper.
- The Abstract was written too broadly. Topics related to the idea of ​​the Natura 2000 should be kept to a minimum. The achieved results should also be described in one/two sentences.
The abstract has been revised to minimize information on Natura 2000 and introduce details on the obtained results.
- The Introduction contains the necessary information about the background of the issue. 46 literature items were analyzed in it. There are 98 of them throughout the article. At the end of this chapter, please clearly define the purpose of the work.
The purpose of the work was better highlighted together with a general shortening of the introduction as also requested by referees 1 and 4.
- The Methodology has been described in a satisfactory and clear manner.
We thank the reviewer for such a positive comment.
- I propose to separate the Results from Discussion as the Discussion has not actually been carried out. It should refer to how the presented results compare to other similar studies. It should be written at a relatively high level of generality.
A specific Discuss section has been added with more general comments.
- The article should be written impersonally avoiding the words "we", "our" etc.
The impersonal form has been adopted throughout the text.
- Please check all units, as in some places there is no space between the number and unit (see for example lines 72 and 81).
The spacing has been checked.
- In References. Why was the title of the journal in item 1 written in capital letters? Please also use abbreviations in relation to the names of journals, in accordance with the editorial recommendations.
The format of references has been implemented in accordance with RS template.
Reviewer 3 Report
The manuscript presents a Smart Procedure for Assessing the Health Status of Terrestrial Habitats in Protected Areas using satellite remote sensing images in GIS bases analysis. The article is good, however, it may require some improvements prior to its possible publication in the Remote Sensing.
The abstract is good, authors could consider briefly describing the satellite data used in the study along with associated methods.
Introduction: The introduction is very good; the authors demonstrate a thorough knowledge of the published literature and highlight the importance and background to carry out this investigation.
L 72: Abbreviation should be explained when appeared first
Study Area: Figure 1, I think it should be revised. The location map should be zoomed out to represent the context of the study area for a global audience and the picture quality should be improved.
Material and Methods: Methods are technically strong and well explained.
Satellite Data
L 226: Reference style, please correct.
L 269: SPOT was launched in 1986, this statement could be reconsidered.
I wonder why the analysis period was not extended to 2020. It would be better to know the current status of the study area, therefore, I strongly recommend extending the analysis till 2020
Results and Discussion.
Figure: 4a could be removed.
Conclusion: NO Comment
Author Response
Reviewer 3
The manuscript presents a Smart Procedure for Assessing the Health Status of Terrestrial Habitats in Protected Areas using satellite remote sensing images in GIS bases analysis. The article is good, however, it may require some improvements prior to its possible publication in the Remote Sensing.
We thank the reviewer for providing suggestions that allowed us to improve our paper.
- The abstract is good, authors could consider briefly describing the satellite data used in the study along with associated methods.
The information has been added.
- Introduction: The introduction is very good; the authors demonstrate a thorough knowledge of the published literature and highlight the importance and background to carry out this investigation.
We thank the reviewer for such a positive comment.
- L 72: Abbreviation should be explained when appeared first
The MODIS acronym has been explained.
- Study Area: Figure 1, I think it should be revised. The location map should be zoomed out to represent the context of the study area for a global audience and the picture quality should be improved.
Figure 1 has been rearranged by contextualizing the study area within Europe (and also combined with figure 2 as suggested by reviewer 4).
- Material and Methods: Methods are technically strong and well explained.
We thank the reviewer for such a positive comment.
Satellite Data
- L 226: Reference style, please correct. https://natura2000.eea.europa.eu/Natura2000/SDF.aspx?site=IT9210210 (last access 30/03/2022).
Since we did not find indications in the instructions for the authors, we used the style adopted in recent publications (last access 30 March 2022) (see, e.g., https://doi.org/10.3390/rs14051180 ; https://doi.org/10.3390/rs14061467 ; https://doi.org/10.3390/rs14051217).
- L 269: SPOT was launched in 1986, this statement could be reconsidered.
The reviewer is right, also SPOT is able to provide suitable historical data; this sentence refers to free downloadable data. It has been finalized. See line 73 of the new version of the manuscript.
- I wonder why the analysis period was not extended to 2020. It would be better to know the current status of the study area, therefore, I strongly recommend extending the analysis till 2020
The analyses presented in this paper are fundamentally dedicated to the validation of the method proposed for assessing the state of habitats. Therefore, they are linked to the availability of data from the in situ surveys. The historical analysis was added to verify some hypothesis of degradation processes indicated in the field reports.
A temporal extension of the work is in any case planned with the conversion of the other sites (28) of the network from SCI to SAC and the final definition of the habitat boundaries.
Results and Discussion.
- Figure: 4a could be removed.
Figure 4a has been removed.
Conclusion: NO Comment
Reviewer 4 Report
The overall theme of the article (monitoring the ecological quality of Natura habitats) is definitely worth investigating and the results about modeling accuracy are very good. Therefore, I feel that the article could be published but the quality of the presentation should be improved a lot.
Abstract:
- please add results (numbers) of how well the model worked
- delete the first sentence
- l12-13: should be "conservation of threatened species"
Introduction:
- the beginning of the section (two first paragraphs) could be shorter. The text here is not very relevant to a remote sensing paper
- the second paragraph is very long and should be divided into multiple paragraphs
- l73: quickbird has a spatial resolution of meters, not tens of meters
- l82-87: to which does this paragraph relate to? Delete or move to another place
- l107: should probably be "outlier" not "outliner"
- the end of the introduction is not very clear, please provide exact research questions and elaborate what is done and where. Now it seems that the research in Basilicata is just a demonstration of the method even though all research was conducted in there
Sections 2, 3 and 4 could be combined into Materials and methods section
- there is two much general text in section 2 and you should just describe your study area concisely and include the text in 3.1 into study area description
- figs 1 and 2 could be combined
- l201-206: this is fine but for me the difference between these two analyses was not very clear in the end
- 3.2: why did you include only two images and did not monitor temporal trends e.g. through Google Earth Engine?
- l309: should probably be "through" instead of "thought"
- l312: this line is probably left here accidentally
- the term "normalized NDVI" is a bit strange as it would without abbreviation "normalized normalized difference vegetation index". Please consider using another term
- could the habitat type tables be combined or would they fit better in the supplementary materials?
results and discussion
- the key results are in 5.3. I would consider to start the result section with this section.
- Please add tables about classification accuracy separately for each site (and possibly also for each habitat type)
- the text on p. 12-25 is exhaustive. The text should be significantly shortened and only the key points should be included. Some of the figures could be in the supplementary material. It would be best to have only one map per study area.
- The quality of the figures is not very good.
- Some of the orthophotos (or are they orthophotos?) are very difficult to interpret due to low contrast. Please add also information about the photos in the captions.
- some of the legends have very poor resolution
- There is no proper discussion (the relevancy of results, reflection of earlier research etc.) at all. Discussion should definitely be added.
- It would have been very good to have a proper change analysis with multiple images, including images in between 1985 and 2009 and after 2009. This analysis would improve the quality of the manuscript but can be left to future research also.
Author Response
Reviewer 4
The overall theme of the article (monitoring the ecological quality of Natura habitats) is definitely worth investigating and the results about modeling accuracy are very good. Therefore, I feel that the article could be published but the quality of the presentation should be improved a lot.
We thank the reviewer for the positive comment. To improve the quality of the presentation, we followed all the suggestions.
- Abstract: please add results (numbers) of how well the model worked; delete the first sentence; l12-13: should be "conservation of threatened species"
The abstract has been modified following the reviewer’s suggestions.
- Introduction: the beginning of the section (two first paragraphs) could be shorter. The text here is not very relevant to a remote sensing paper. The second paragraph is very long and should be divided into multiple paragraphs
The introduction has been revised and shortened.
- l73: quickbird has a spatial resolution of meters, not tens of meters
The information has been corrected (line 52 of the new version of the manuscript).
- l82-87: to which does this paragraph relate to? Delete or move to another place
The sentence has been rearranged.
- l107: should probably be "outlier" not "outliner"
The misprint has been corrected.
- The end of the introduction is not very clear, please provide exact research questions and elaborate what is done and where. Now it seems that the research in Basilicata is just a demonstration of the method even though all research was conducted in there
The referee is right, the analyses presented in this paper are fundamentally dedicated to the validation of the method through the comparison with data from detailed field surveys. Anyway, the research question has been better highlighted in the revised text.
- Sections 2, 3 and 4 could be combined into Materials and methods section
The sections have been combined following the reviewer’s suggestion.
- There is two much general text in section 2 and you should just describe your study area concisely and include the text in 3.1 into study area description
The study area section has been shortened and moved as sub-section of Materials and Methods.
- figs 1 and 2 could be combined
The figures have been combined.
- l201-206: this is fine but for me the difference between these two analyses was not very clear in the end.
By calculating the statistics at the SAC and network level, the status of each habitat is assessed within the single protected site and with respect to all the sites in the network that contain that particular habitat. Basically, from a management point of view, with these two levels of analysis, the method can support local managing bodies and the regional authority by identifying priority areas at site (local) or network (regional) level. We have better clarified this concept.
- 2: why did you include only two images and did not monitor temporal trends e.g. through Google Earth Engine?
We implemented such a procedure to avoid a direct estimation of trends on a spectral index. Due to the discrete temporal sampling of Landsat data (16 days), trends can be biased by little shifts in phenology induced by different acquisition dates of cloud-free imagery or by the peculiar meteo-climatic condition of a given year. Moreover, in patchy habitats (e.g., 6210* in Monti Li Foi SAC, 6420 in Monte Vulture SAC) a pixel-based trend analysis can result in a salt-pepper map of trends when the dimension of the patches with higher vegetation cover (respect to the herbaceous background) is comparable with Landsat pixel size and collocation error.
Tools in GEE, such as LandTrendr, can be a very useful support to identify drastic changes than cannot be obscured by shift in phenology and to analyze macro-habitats (e.g., forest, grassland, shrub).
The main objective of the proposed method is the identification of incipient (and therefore subtle) modifications of the habitats that emerge as anomalies with respect to the standard habitat conditions.
In a temporal domain, we can look for persistence or changes of anomalous areas.
- l309: should probably be "through" instead of "thought"
The misprint has been corrected.
- l312: this line is probably left here accidentally
The reviewer is right; the line has been removed.
- The term "normalized NDVI" is a bit strange as it would without abbreviation "normalized normalized difference vegetation index". Please consider using another term
The term has been substituted with standardized.
- could the habitat type tables be combined or would they fit better in the supplementary materials?
As the habitat codes and names can be keywords for the ecology community, the tables have not been moved to supplementary materials in order to not limit web searches.
However, we have combined them in a single table (Table 1 of the new manuscript).
results and discussion
- The key results are in 5.3. I would consider to start the result section with this section.
The Accuracy section has been moved as first subsection of Results.
- Please add tables about classification accuracy separately for each site (and possibly also for each habitat type)
We reported the accuracy for each site in Table 2.
- The text on p. 12-25 is exhaustive. The text should be significantly shortened and only the key points should be included. Some of the figures could be in the supplementary material. It would be best to have only one map per study area.
To shorten this section, results for 1985 (pp.19-25) have been moved in the supplementary material.
- The quality of the figures is not very good. Some of the orthophotos (or are they orthophotos?) are very difficult to interpret due to low contrast. Please add also information about the photos in the captions. Some of the legends have very poor resolution.
The referee is right: they are orthophotos; the information has been added in the captions.
We apologize for the poor quality of the images but it is an effect of pdf conversion. Anyway, we increased the resolution to minimize the effects of conversion.
However, as for the first submission, we have also sent all the figures at full resolution in tif format which should be accessible via the MDPI platform or alternatively via links to be requested from the editor.
- There is no proper discussion (the relevancy of results, reflection of earlier research etc.) at all. Discussion should definitely be added.
We thank the reviewer for the suggestion; a discussion section has been added.
- It would have been very good to have a proper change analysis with multiple images, including images in between 1985 and 2009 and after 2009. This analysis would improve the quality of the manuscript but can be left to future research also.
We agree with the reviewer, a temporal extension of the work is planned with the inclusion of all the other sites (28) of the network in conversion from SCI to SAC and subject to a final boundary definition.
However, the analyses presented in this paper are fundamentally dedicated to the validation of the proposed method with 2009 in situ surveys. The historical analysis was added to verify some hypothesis of degradation processes indicated in the field reports.
Round 2
Reviewer 2 Report
Please remove from the title of the chapter 5 a word "Discussion" because it is added the next one - no 6.
Please also remove from the "Conclusions" all citations.
After that the article could be published.
Author Response
Reviewer 2
Please remove from the title of the chapter 5 a word "Discussion" because it is added to the next one - no 6.
The title of Section 5 has been changed in “Results” and the “Discussion” section has become section 6.
Please also remove from the "Conclusions" all citations.
Citations have been moved from the Conclusions to the Discussion section.
After that the article could be published.
We thank again the reviewer for providing suggestions that allowed us to improve our paper.
Reviewer 3 Report
I am happy with the revised version.
Author Response
Reviewer 3
I am happy with the revised version.
We thank the reviewer for the positive comments and for providing suggestions that allowed us to improve our paper.
Reviewer 4 Report
-
Author Response
Reviewer 4
We thank again the reviewer for providing suggestions that allowed us to improve our paper.